# Development of the Phaseless Calibration Algorithm for a Digital Antenna Array

Elena Dobychina, Mikhail Snastin, Vladimir Savchenko and Timofey Shevgunov *

Moscow Aviation Institute, Volokolamskoe Shosse 4, Moscow 125993, Russia; dem5577@gmail.com (E.D.);
mexanizmys@ya.ru (M.S.); vnsavchenko@mail.ru (V.S.)
* Correspondence: shevgunov@gmail.com

**Abstract:** In this paper, we will discuss a calibration algorithm for a digital antenna array that diagnoses its real performance. It can be applied at such stages of the antenna system life cycle as design, tuning, and especially maintenance. A calibration implementation using a scalar method for a multi-beam digital antenna array is proposed and investigated. On-the-fly calibration ensures a continuous improvement in beam pointing accuracy by reducing internal errors in the receiving (transmitting) channels. The purpose of the study is to experimentally examine the capabilities of digital beamforming to increase the angle-of-arrival estimation accuracy. A simulation model of the receiving antenna was created in an anechoic chamber with a planar antenna positioner. The possibility of precise estimation of the direction of arrival using the digital beamforming with electronic scanning was demonstrated. The proposed simulation model made it possible to observe the convergence of the antenna array calibration process using the proposed method for various errors in the signal paths, as well as different signal-to-noise ratios. It has been proven that even under adverse conditions early in the calibration algorithm, the phase error detection converges with high accuracy, and its value decreases uniformly even to the fractions of an angular degree.

**Keywords:** calibration algorithm; digital antenna array; simulation model; anechoic chamber; method convergence; phase error

## 1. Introduction

Unmanned aerial vehicles (UAVs) are now in demand in all areas of activity. The UAV must have a radio-electronic system (RES) on board to implement control, navigation, and communication tasks. The latest RES requires multi-functionality and high disturbance immunity [1,2]. This can be achieved by moving from phased (i.e., analog) antenna arrays (PAAs) to digital antenna arrays (DAAs). Such an array is based on digital transceiver modules (digital TRM, DTRM) [3]. The DAA beam steering requires highly accurate control of signal parameters, so knowledge of the actual gain and phase shift across the DTRM is necessary. To provide the benefits of digital beamforming, the characteristics of transmit and receive channel components must be highly identical. However, electronic components change over time and temperatures drift, and parameters of the on-board devices are also subject to mechanical influences. As a result, the amplitudes and phases at the output of different channels differ from the specified values, which leads to errors in the amplitude-phase distribution (APD) across the DAA aperture and then to worsening in such important essences as the directivity, efficiency, and sidelobe level (SLL).

Antenna array diagnostics, i.e., the process of monitoring the state of actual APD and faults, is one of the most important tasks in the process of development and configuration, and especially during the operation of the DAA. Currently known methods for antenna diagnosing differ in their implementation approaches. These methods are divided into low frequency and high frequency. Low-frequency methods consist of monitoring phase shifter control circuits, matching digital codes and analog beam steering control signals [4,5].

The disadvantage of these methods is the lack of actual APD in the operating frequency band over the aperture. High-frequency diagnostic methods make it possible to estimate each channel error, i.e., to evaluate APD over the aperture. High-frequency diagnosis methods are classified according to various criteria. One of the classifications divides these algorithms according to the place of implementation into bench methods and regular ones, i.e., for diagnosing on-board antenna systems as part of the RES. Bench (or external) methods are used during PAA developmental tests where the performance of the beam steering system is monitored. Such diagnostic methods are based on measuring and analyzing the parameters of the PAA field in the near- or far-field regions, which may require large-sized antenna test ranges. Basically, these methods are implemented in anechoic chambers (AECs) using a set of measuring equipment and a test antenna [6–10]. APD measurement is the first stage of the procedure for restoring antenna characteristics using near-field methods [7]. Vector near-field (holographic) methods are based on direct measurement of the electromagnetic field near the antenna aperture using a scanning probe or multi-element antenna measurement tool [11]. These measurements can also be taken using compact antenna test ranges with a special reflector that helps overcome the large far-field distance criteria.

Internal diagnostics of the RES antenna system is usually carried out as part of startup checks to monitor its performance or during routine maintenance. Regular methods use built-in equipment and internal signals of the RES transmitter itself (internal methods) [12]. A common feature of internal methods is the embedding of additional elements into the PAA exclusively for the purpose of a diagnosis—an integrated control system.

The second way to classify high-frequency methods is to divide them into vector and scalar, or phaseless, ones. To implement phase methods, it is necessary to ensure phase synchronization of the measuring equipment (measuring receiver, signal generator, etc.) and the antenna under testing, i.e., to have a reference channel [13,14]. The output signal of the sum channel, the output signal of the single channel, or the source of the transmitter signal for two-way (transmitting and receiving) antennas can be selected as a reference.

The above classification of diagnostic methods is very arbitrary. Methods that fall into one group or another according to some criteria may be quite feasible in other groups of methods. For example, phase methods can be implemented both in bench diagnostics of antenna arrays and in regular diagnostics of the RES, and holographic methods can also be scalar, using only amplitude distribution over multiple scan planes [15,16].

Antenna diagnostic results can be useful in solving the following:

- compensation for distortions introduced by faults in the antenna array during the RES operation;
- adjustment of the APD in the controlled array channels in order to maximize the desired performance of the RES. Uncontrolled failed channels are subject to repair and replacement;
- recording of a real antenna APD in the processor memory for later use with modern signal processing methods into the RES [17].

The possibility and quality of solving these problems are determined by the diagnostic method used, which largely depend on the composition and design of the RES [17]. The results of periodic or ongoing antenna diagnostics must be taken into account when operating the entire system. In case of continuous operation, i.e., with each new phasing of the array, and its results are taken into account automatically, then adaptive RES becomes the subject of discussion [18]. With the development of circuit components, interest in millimeter-wave radars arose [19,20]. Their significant advantage was the reduction in size; an "antenna-on-chip" and an "antenna-in-package" appeared [21–26]. The progress of modern technologies has made it possible to develop this frequency band [27,28]. However, calibration issues for millimeter-wave antenna arrays are even more relevant due to the drifts in channel parameters.

Calibration of antenna arrays is in demand in complexes of various purposes. These are multi-user MIMO systems [29,30]; various on-board aviation systems [31,32]; ground

and satellite stations [33–36]; and even floating [37] or rotational platforms [38]. Calibration algorithms are constantly being improved, new ones are appearing, and known methods are being experimentally studied [39–41]. The brief summary of the main methods exploited for performing calibration is presented in Table 1.

**Table 1.** Comparison of calibration methods.

| Frequency Bands | Method | Classification Attribute | Characteristic Feature |
|---|---|---|---|
| High frequency | Bench | testing of antenna array | requires test facility, may be realized in an anechoic chamber |
| | Regular | routine maintenance | use of internal digital signals with the integrated control system |
| | Near-field | measurement of the field distribution near the aperture | implemented with a near-field scanner |
| | Far-field | measurement at the far-field distance | requires special antenna test ranges |
| | Vector | reference signal is required | use of high-precision measuring equipment |
| | Scalar (phaseless) | no special phase reference is needed | amplitude-only measurement |
| | External | analysis of the field distribution over the aperture or near- or far-field regions | near-field measurements are most promising |
| | Internal | requires additional elements in the PAA | autonomous calibration possible, requires internal control system |
| | REV | measurements in all possible phase states | the large number of measurements required to achieve accuracy |
| | Current method | measurements in four orthogonal phase states | reduced calibration time |
| Low frequency | | monitoring the integrity and serviceability of control circuits, matching digital codes and analog beam steering control signals | does not provide information about the real amplitude and phase distribution of the antenna array |

It is necessary to highlight the antenna calibration based on the rotating element electric field vector (REV) method [42]. These are the most common algorithms when the sum signal of all array channels, except the one being calibrated, is used as a reference [43–48]. Their significant drawback is the large number of measurements required to achieve acceptable accuracy.

Currently, antenna arrays with digital beamforming are increasingly used in radio systems for various purposes [49–51]. There are obvious advances in the theory of digital signal processing (DSP) [52,53], as well as in the production of modern circuit components [54–57].

In the DAA, it becomes possible to diagnose and adjust the APD, i.e., to carry out high-precision calibration during the operation of the antenna and the entire RES as a whole. This is especially in demand in on-board multi-functional systems, when it is necessary to carry out in-flight calibration.

This paper presents a calibration method, which is a development of the phaseless method for the case of a digital antenna array. The novelty of the work lies in the transfer of experience in PAA calibration in relation to digital antennas. The purpose of the work is to study a simulation model of the proposed calibration method in relation to on-board multi-beam DAA. In what follows, a simulation model will be understood as a physical model of an antenna array obtained using a single movable emitter. The number of analog devices that introduce uncontrolled random amplitude and phase errors into beamforming was reduced to a minimum. It is shown that such a solution increases the pointing accuracy of each DAA beam to a fraction of a degree.

## 2. Materials and Methods

### 2.1. An Implementation of Antenna Array Calibration

For on-board DAA of the UAV, a phased array calibration approach [58] can be applied. When calibrating an analog multi-beam antenna, one of the beams is aimed at a fixed source of the calibration signal, which allows the control of individual channels. During the calibration of a single channel, the phases of the remaining channels involved in the beamforming are fixed and equal to the values necessary to point at the source of the calibration signal. Subsequently, the phase of the calibrated channel changes by 90°, 180°, and 270° relative to the initial state using a controlled phase shifter. The phase and amplitude errors in this channel are related to the power measured in the four phase states. The procedure is performed for each channel and cycling through four orthogonal phases reduces calibration time.

The disadvantage of this analog case is that its measurement system contains a narrow-band filter and a power detector whose output voltage is proportional to the power of the received radio signal. The phase of the calibrated channel is changed with controlled phase shifters. These devices are also analog and change their characteristics over time, as well as under the influence of various external conditions such as ambient temperature or power supply instability. This leads to the presence of random amplitude and phase errors within the calibration system itself, which reduce the accuracy of determining the corrections, and, consequently, the accuracy of beam steering.

Therefore, it is proposed to investigate a calibration using the same idea, but in the context of the multibeam DAA, which improves the beam pointing accuracy by reducing errors in the TRM channels. A simplified block diagram of this approach is presented in Figure 1.

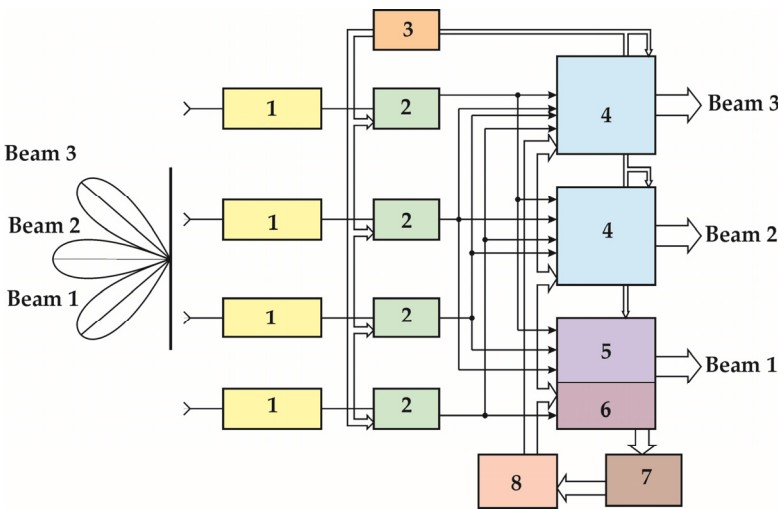

**Figure 1.** DAA structure with integrated calibration channel.

The figure shows a diagram of a calibration channel as part of a multi-beam DAA, where the following applies: 1—an analog microwave part of one antenna receiving (and/or transmitting) channel; 2—an analog-to-digital converter (ADC) of the receiving and digital-to-analog converter (DAC) of the transmitting antenna channel, respectively; 3—the source of the digital synchronization signal; 4—DSP for processing a single DAA beam; 5—DSP for processing a calibrated beam (DSPC); 6—the calibration unit as part of the DSPC; 7—the computing unit of corrections as part of the calibration unit; 8—the computing unit of amplitude and phase coefficients as part of the calibration unit.

In both the receiving and transmitting paths of the DAA, in principle, there are no controlled phase shifters and analog attenuators. Thus, calibration is carried out with digital methods in the DSP (DSPC 5 in Figure 1 in our case), where calibration coefficients

are calculated. The phase in the channel being calibrated is periodically changed, but digitally rather than analogue-wise. For example, to calculate the received power with DSPC, you can provide sampling with time intervals proportional to the introduced phase shifts. Since all DAA channels are involved in the formation of each beam, single-beam calibration allows you to adjust the amplitude and phase coefficients of each channel. Once calibration adjustments are obtained (e.g., amplitude and phase coefficients of all channels), this information is fed to the corresponding DSP, which applies this correction in the beamforming process. Additional units perform all the necessary functions as part of DAA channels, while eliminating a number of analog devices, such as attenuators, controlled phase shifters, a narrow-band filter, and a quadrature detector, which introduce uncontrolled errors into the radiation pattern synthesis. Thus, the pointing accuracy of each DAA beam is increased.

Additional units are common elements of digital antennas, which make it possible to reproduce the proposed calibration structure in full.

To meet the challenge of studying the digital array calibration algorithm, it is necessary to first reproduce the digital beamforming process using an experimental model of a receiving linear array with equidistant emitters.

### 2.2. Experimental Setup and Simulation Results of Digital Beamforming

Digital antenna arrays, despite their growing use, are still an expensive product. Therefore, we will use a simulation model of such an antenna, reproduced in an anechoic chamber [59]. To examine the capabilities of digital beamforming, an experimental setup was assembled that allows for reproducing the direction finding process using DAA and estimating the angular position of the target simulator. The measurement setup is shown in Figure 2a. During the research, discrete spatial measurements of the complex transmission coefficient between the emitting antennas Satimo SH800, QR2000, SH4000: MVG (SATIMO), Villebon-sur-Yvette, France (see Figure 2b) and the elements of the antenna array were carried out with a vector network analyzer (VNA) Keysight PNA N5225A: Keysight (Agilent), Santa Rosa, CA, U.S. The linear array was simulated using a single measuring probe (see Figure 2c) mounted on a planar near-field scanner [60]: VEMZ, Dubna, Moscow region, Russian Federation, by X-axis movement as shown in Figure 2d. The digital signal processing, numerical evaluation and visualization of the obtained results were conducted by means of Scilab: Dassault Systèmes, Paris, France.

First, measurements were carried out at a 3 GHz carrier frequency, the VNA resolution bandwidth (RBW) set to values of 1 kHz and 10 kHz to simulate different signal-to-noise ratios (SNRs). The transmitting antenna as a target simulator was first installed normal to the central position of the receiving aperture emitter, i.e., the middle position of the antenna on the scanner axis. The complex transmission coefficient was measured between two co-polarized antennas depending on the spatial position of the receiving antenna along the X-axis. The measurement was conducted for several power levels of the transmitter and different RBW of the receiver with vertical polarized antennas.

In the beginning, the receiving antenna moved in steps equal to half the wavelength, covering a distance of 0.4 m. The target emitter was located in the far-field of the virtual (simulated) receiving antenna array. The far-field distance for an antenna array with similar parameters is 3.2 m, and the target antenna was located at a distance of at least 3.25 m away from the scanner during the measurements.

As a result of the measurements, vector samples were obtained containing the distributions of amplitude ($A$) and phase ($\varphi$) along the plane of the virtual linear array, presented in Figure 3 and Figure 4, respectively. On the graphs, **X** corresponds to the position offset of the receiving antenna on the scanner relative to the normal (middle) location.

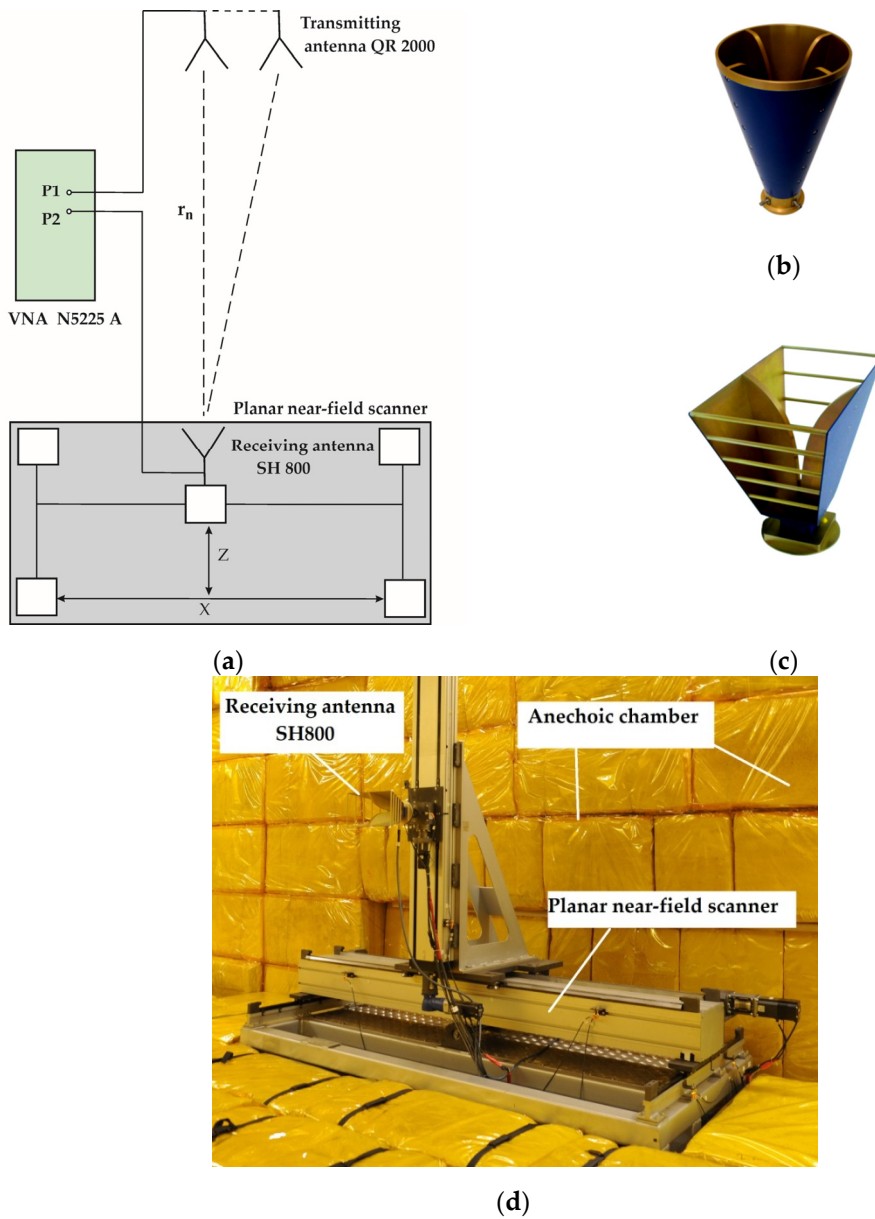

**Figure 2.** Measurement setup in the AEC: (**a**) Simplified measurement diagram; (**b**) Satimo QR2000 test antenna; (**c**) Satimo SH800 test antenna; (**d**) Planar near-field scanner with receiving antenna installed.

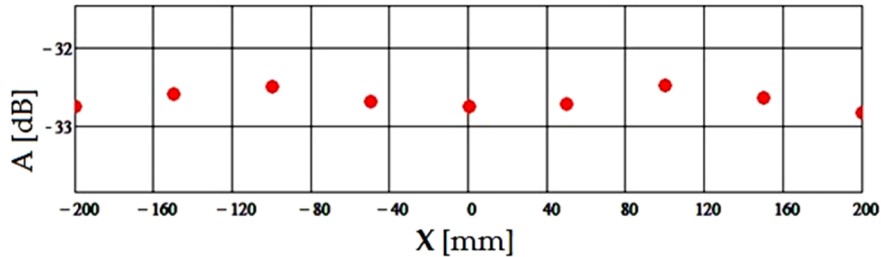

**Figure 3.** Amplitude distribution of the simulated DAA.

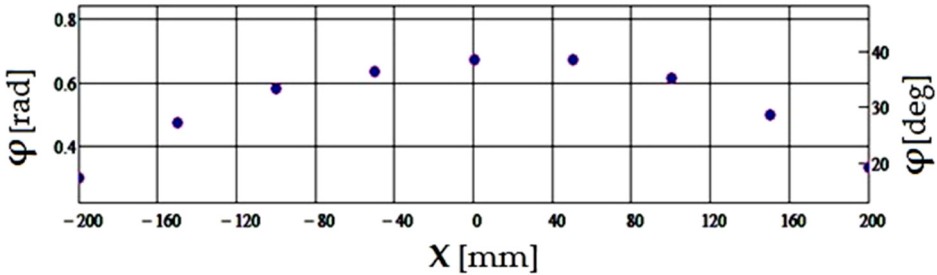

**Figure 4.** Phase distribution of the simulated DAA.

Appropriate software was developed in accordance with the composed algorithm for estimating the direction of arrival (DoA). To do this, we first reconstructed the complex signal from measured samples of the amplitude and phase of the received signal for each element position of the virtual antenna.

$$s_n = A_n \exp(j\varphi_n), \tag{1}$$

where $A_n$ and $\varphi_n$ are, respectively, the amplitude and phase samples of the received signal for the $n$-th antenna position of the simulated antenna array. Next, a fast convolution algorithm [61,62] of the received signal $s_n$ with signals $s_{et}(\Theta_i)$ obtained from all possible directions $\theta_i$ using the simple transmission equation is applied:

$$s_n * s_{et}(\Theta_i) = \frac{1}{N} \sum \text{IDFT}\left[(\text{DFT}(s_n))^* \cdot (\text{DFT}(s_{et}(\Theta_i)))\right], \tag{2}$$

where DFT and IDFT stand for forward and inverse discrete Fourier transform, respectively; $x^* = \text{Re}x - j \cdot \text{Im}x$ denotes complex conjugation; $N = 9$ is the number of elements of the simulated DAA, i.e., the number of receiving antenna positions on the scanner during the experiment. The calculated cross-correlation function (CCF) for a set of spatial angles from the range $\Theta = -\pi/2 \ldots \pi/2$ with a step (resolution) $\delta\Theta_i = 1°$ is presented in Figure 5 in the form of a heat map, where the highest value is shown as a lighter color (i.e., white).

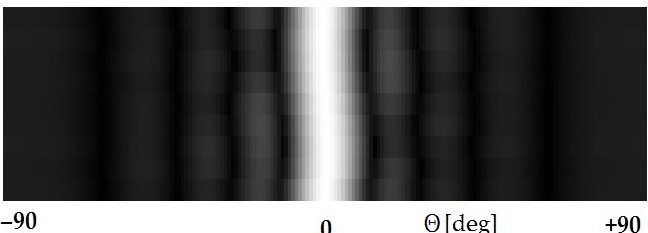

**Figure 5.** The CCF heat map for simulated DAA.

Based on the results of the CCF, it is possible to reconstruct the digital radiation pattern (DRP) of the simulated antenna array (see Figure 6a), the maximum of which corresponds to the DoA of the target simulator signal. A polar plot of the DRP is shown in Figure 6b.

The resulting 3 dB beamwidth of the DRP is 11.4°. This value corresponds to the beamwidth of a single emitter taking into account the array factor for a simulated antenna array. Figure 6 shows that the position of the maximum of the digitally synthesized radiation pattern corresponds to the 0° of the incidence, which demonstrates the possibility of direction finding using digital processing. The obtained sidelobe level equals to −12.7 dB, which corresponds to a given uniform amplitude distribution. It should be noted that the experiment was carried out under clutter conditions, which was reproduced in the AEC via scattering sources—for example, various metal parts of the near-field scanner. Naturally, this noise is significantly lower than in a real radar situation on a UAV board; in addition, it is necessary to keep in mind the possibility of active jamming coming from the

target. Figure 7 shows the synthesized difference radiation pattern obtained using digital processing.

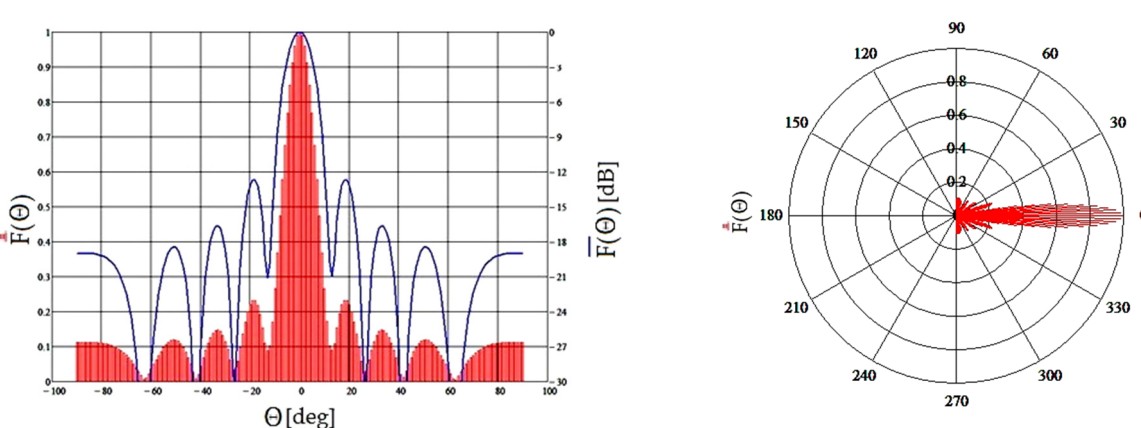

(a)
(b)

**Figure 6.** Digital radiation pattern of the simulated DAA, angle of incidence $0°$: (**a**) Cartesian plot; (**b**) Polar plot.

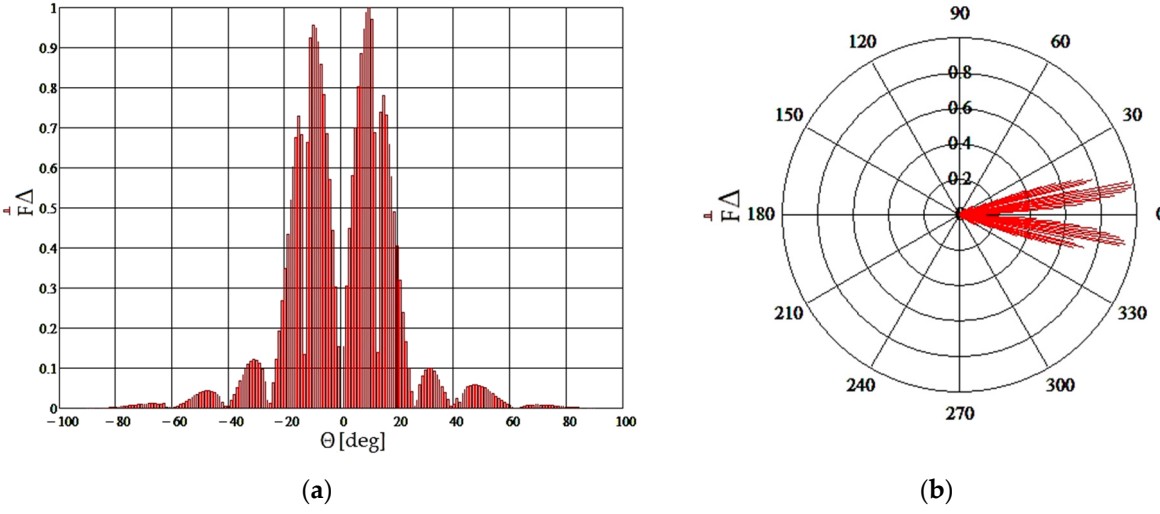

(a)
(b)

**Figure 7.** The difference radiation pattern of the simulated DAA, angle of incidence $0°$: (**a**) Cartesian plot; (**b**) Polar plot.

Digital synthesis of the radiation pattern allows you to increase the accuracy of determining the DoA without additional means, only by changing the parameters of the computation algorithm. An example of constructing a DRP with a finer angular step is shown in Figures 8 and 9. The sum radiation pattern does not provide accurate information about the target angular position due to the flatness of its curve, and Figure 8 shows that the corresponding direction finding accuracy is $\pm 1°$, while the difference pattern presented in Figure 9a shows an angle estimation accuracy of $\pm 0.1°$. If computational costs are not a limitation for a particular DAA, the error can theoretically be reduced to negligibly small values. This is illustrated by the difference pattern (see Figure 9b), where the angular resolution is $0.001°$.

To demonstrate the capabilities of direction finding using a virtual DAA prototype, the experiment was repeated for a new target position. The transmitting antenna was moved approximately 1.5 m away from the center of the simulated receiving antenna array, i.e., the central position of the antenna on a planar scanner. This deviation at a given distance corresponded to a target angular movement of approximately $25°$ from

the normal to the simulated antenna aperture. The angular position was measured using coarse means available and a more accurate measurement was assumed using the DAA prototype. The complex transmission coefficient between the two co-polarized antennas was then re-measured as the X-axis position of the receiving antenna changed. As before, the carrier frequency was 3 GHz, and the receiving antenna moved in steps equal to half the wavelength on the scanner (see Figure 2d), covering a distance of 0.4 m.

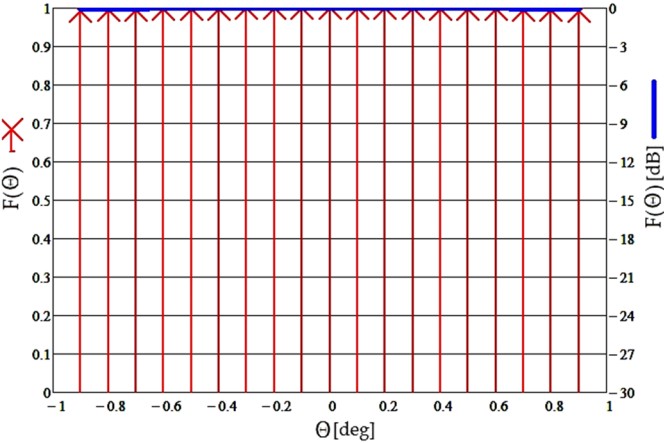

**Figure 8.** The sum radiation pattern of the simulated DAA with an angular resolution of 0.1°.

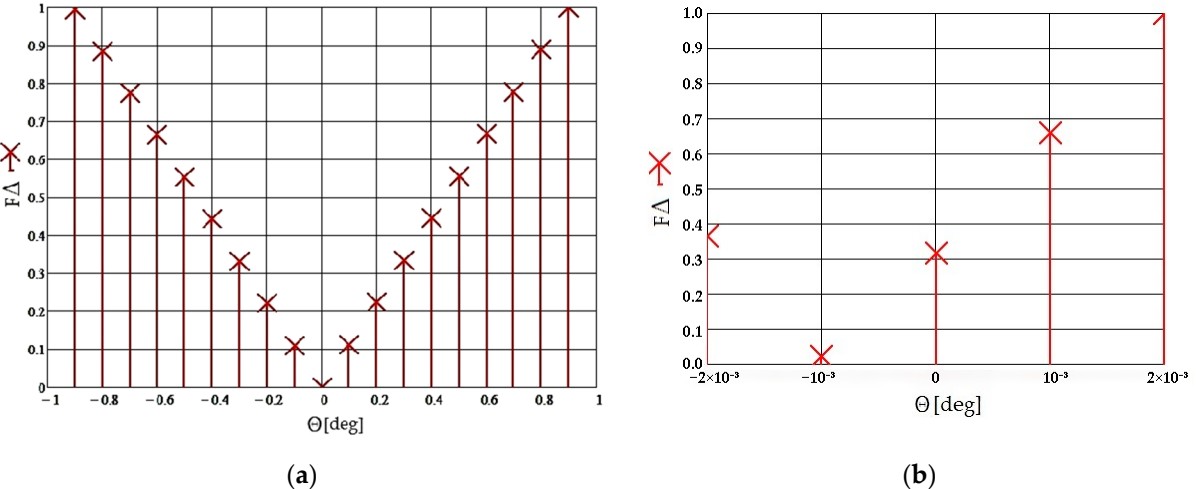

(**a**)                                   (**b**)

**Figure 9.** The difference radiation pattern of the simulated DAA with an angular resolution of (**a**) 0.1°; (**b**) 0.001°.

As a result, new values of the amplitude and phase distribution of the simulated DAA were obtained, which are shown in Figure 10 and Figure 11, respectively.

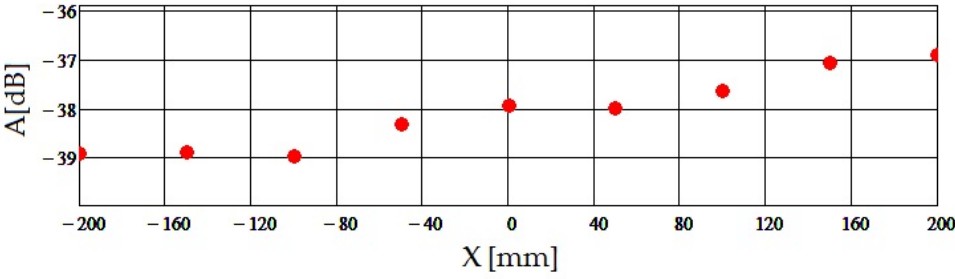

**Figure 10.** Amplitude distribution of the simulated DAA, angle of incidence approx. 25°.

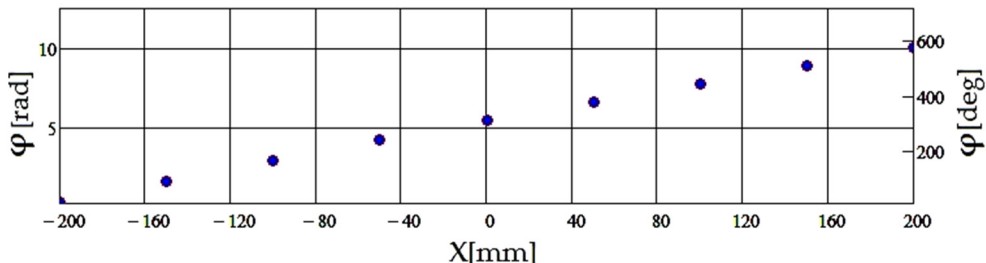

**Figure 11.** Phase distribution of the simulated DAA, angle of incidence approx. 25°.

The CCF for a set of spatial angles from the range $\theta = -^\pi/_2 \ldots {}^\pi/_2$ with a step $\delta\theta_i = 1°$ is presented in Figure 12. It can be seen that the highest correlation (lighter color) has noticeably shifted in the direction away from 0°.

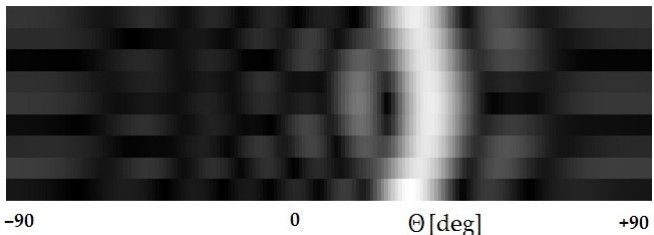

**Figure 12.** The CCF heat map for simulated DAA, angle of incidence approx. 25°.

Based on the CCF results, the DRP of the simulated antenna is once again reconstructed (see Figure 13a), the maximum of which corresponds to the DoA of the signal from the target simulator antenna. The DRP in polar coordinates is shown in Figure 13b.

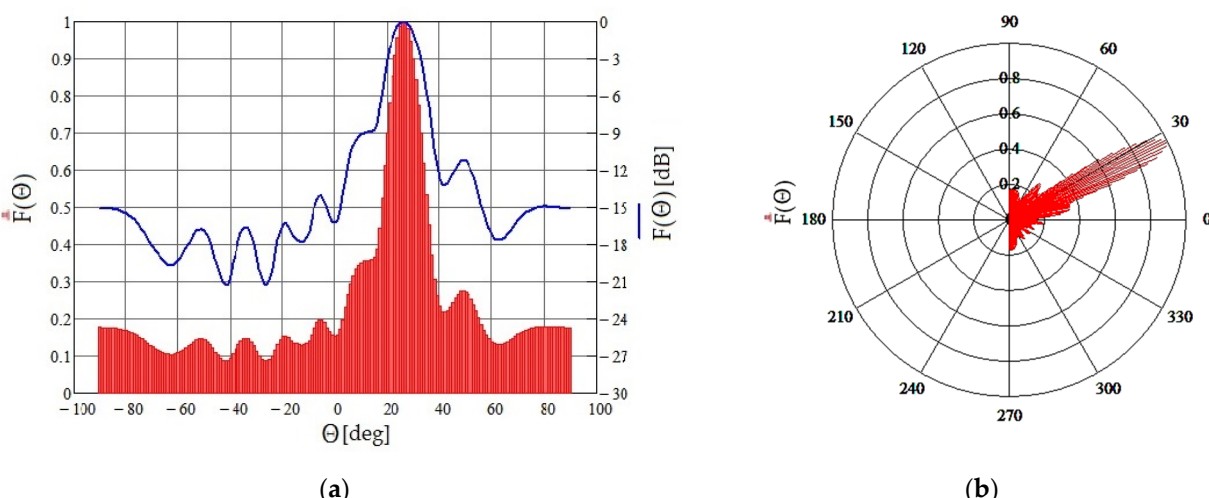

| (a) | (b) |

**Figure 13.** Radiation pattern of the simulated DAA, angle of incidence approx. 25°: (a) Cartesian plot; (b) Polar plot.

The calculation was carried out with an angular accuracy of 1°, so the direction to the target is estimated as 26° ± 1°. The 3 dB beamwidth was increased to $2\theta_{0,7} = 13°$, while the radiation pattern shape became asymmetrical, and the sidelobe levels changed to −9 dB and −12 dB.

To increase the accuracy of estimating the DoA, DRPs with a smaller angular step were calculated. Consider the difference pattern shown in Figure 14a. This allows estimation with an accuracy of ±0.1°; in the example given, it is 26.4° ± 0.1°. The difference pattern (see Figure 14b) with an angular resolution of 0.001° made it possible to detect the angular deviation of the target simulator from the normal by 26.422° ± 0.001°.

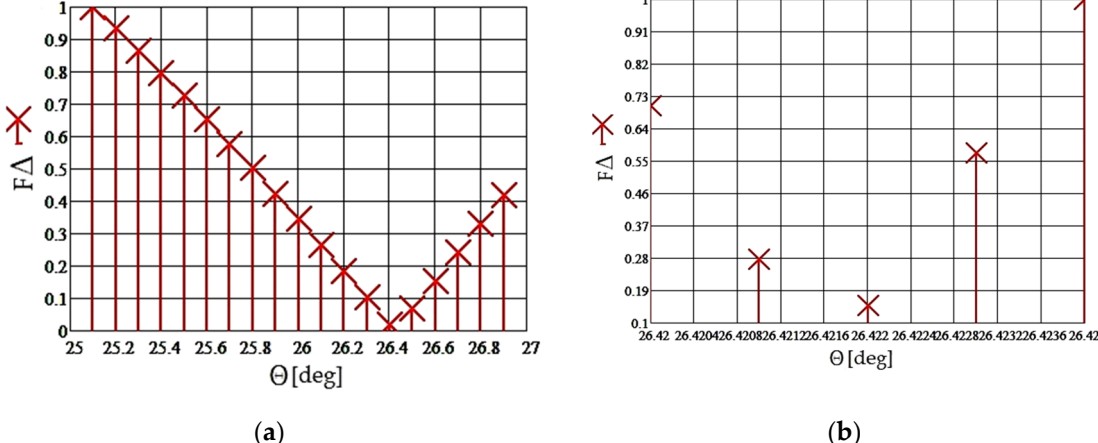

(**a**)    (**b**)

**Figure 14.** The difference radiation pattern of the simulated DAA: (**a**) Angular resolution of 0.1°, DoA is 26.4°; (**b**) Angular resolution of 0.001° DoA is 26.422°.

The experiment was repeated at a 10 GHz carrier frequency using a different set of test antennas (Satimo SH4000 as receiving antenna). The experimental setup is shown in Figure 15. The receiving antenna moved along the X-axis of the scanner with a step equal to half the wavelength, that is, 15 mm, covering a distance of 0.21 m, taking a total of 15 discrete positions. The far-field distance for an antenna array with similar parameters is 2.94 m, and the target antenna was located at a distance of at least 3.25 m away from the scanner during the measurements.

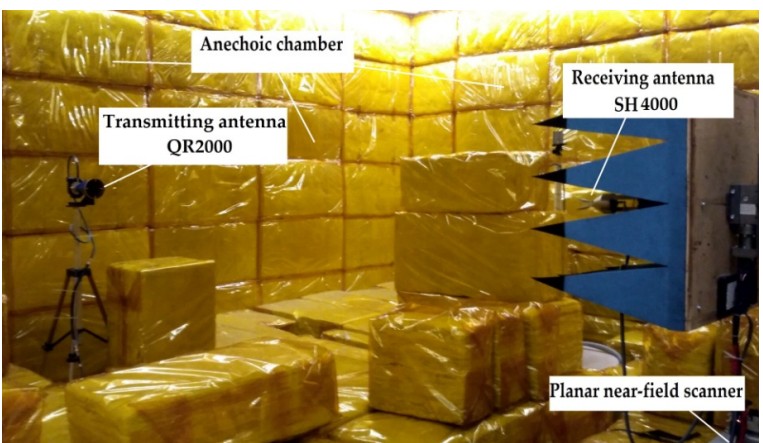

**Figure 15.** Measurement setup at 10 GHz carrier frequency.

Based on the CCF results, the DRP of the simulated antenna array was reconstructed (see Figure 16), the maximum of which corresponds to the DoA for the target simulator.

As expected, a 3 dB beamwidth of the constructed DRP was reduced to 4° compared to operation at a lower frequency. This value corresponds to the beamwidth of a single emitter taking into account the array factor for a simulated antenna array. The SLL corresponds to a given uniform amplitude distribution.

Thus, during the experiment, an algorithm was composed and an example of constructing a digital radiation pattern was shown using a full-scale DAA simulation model. The possibility of precise DoA estimation using the digital electronic scanning has been demonstrated. The measurements were carried out in the presence of clutter, which occurs even in anechoic chambers due to the finite absorption of incident waves (which is associated with the AEC reflectivity level), intentionally present in the experiment. The influence of clutter on the accuracy of DAA parameters is discussed in the next section.

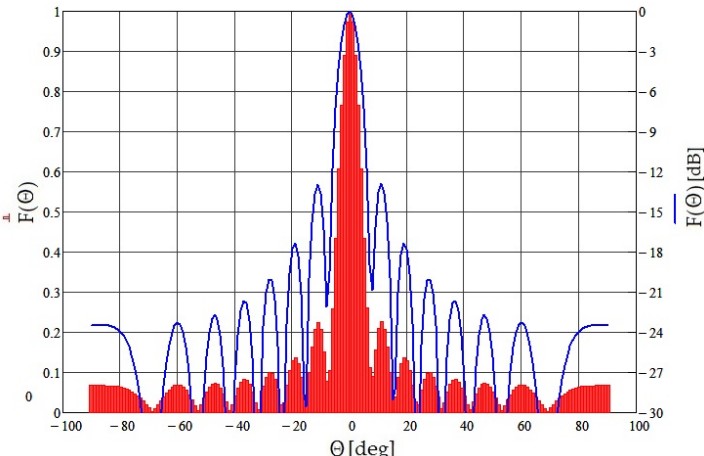

**Figure 16.** Radiation pattern of the simulated DAA at 10 GHz carrier frequency, angle of incidence $0°$.

## 3. Results

### 3.1. Simulation Model of the Calibration System

The implementation of the calibration proposed in Section 2 for on-board multi-beam digital antenna arrays uses a phase-switching scalar method to diagnose amplitude and phase errors in the antenna array aperture [16]. To estimate errors, we can use a simplified element-by-element absolute calibration method, where the phase and amplitude errors in each DAA channel are related to the total power at the output of the entire array measured in four orthogonal phase states [63].

The estimates of the amplitude $\hat{A}_k$ and phase $\hat{\delta}_k$ errors in the calibrated channel are found from the statistical analysis [58,63] of the received signal from the calibration signal source:

$$\hat{A}_k = \frac{\sqrt{\left(\overline{P_{270}} - \overline{P_{90}}\right)^2 + \left(\overline{P_0} - \overline{P_{180}}\right)^2}}{4A_c} \tag{3}$$

$$\hat{\delta}_k = \tan^{-1}\left(\frac{\overline{P_{270}} - \overline{P_{90}}}{\overline{P_0} - \overline{P_{180}}}\right) \tag{4}$$

The random value of $\overline{P_\Phi}$ is the average total power of the received signal calculated over M samples. The phase shift $\Phi$ sequentially takes values of $0°$, $90°$, $180°$, and $270°$ in the calibrated channel of the array. The value $A_c$ is calculated as the sum of signals of all DAA channels except the calibrated one.

The variance of the amplitude and phase estimates are obtained [58,63] from first-order derivatives for the uniform amplitude distribution over the antenna aperture:

$$\hat{\sigma}_A^2 = \frac{N_0 B}{2M}\left(1 + \frac{N_0 B}{2P_k(N-1)^2}\right) \tag{5}$$

$$\hat{\sigma}_\delta^2 = \frac{N_0 B}{4MP_k}\left(1 + \frac{N_0 B}{2P_k(N-1)^2}\right) \tag{6}$$

where $P_k = A_k^2/2$ is the signal power in the calibrated channel; $N_0/2$ is the power spectral density of narrowband Gaussian noise within the bandwidth *2B* of the receiving array channel filter; $N$ is the number of channels (elements) in the array.

RMS calibration error estimates depend on the SNR $N_0 B/P_k$ of the calibrated channel. Multiple sampling during this procedure, averaging, and statistical calculations are used to increase the SNR, i.e., to minimize the influence of the noise.

The phase and amplitude estimates in (3) and (4) imply perfect accuracy in controlling the signal parameters in the calibrated receive or transmit channel during the calibration

process. In a real DAA, quadrature modulators with DACs in the transmit path will not be able to provide an accurate phase setting and will not allow perfect control of the amplitude $A_k$. In the receiving path, the ADC will also introduce amplitude quantization error and phase error due to unavoidable jitter and quantization noise. There will be tuning errors, which have specific values and can also be taken into account in the calibration process.

It is possible to get closer to solving a problem with a given accuracy through an iterative approach. Let us use expression (3) to estimate the initial amplitude error. Expression (4) can be used to estimate the initial phase error and then to iteratively find solutions. Iterations continue until the new estimate converges with the given accuracy. The procedure may be repeated to achieve the desired calibration accuracy. Using a power calculation in each of the four orthogonal phase states of the calibrated element reduces the calibration time, and the use of the maximum likelihood algorithm should ensure convergence of the method.

A simulation model has been developed to observe the convergence of the antenna array calibration process using the proposed method for different errors in the paths as well as different SNR. The developed program can use the initial data specified in the form of a random process, and also use the results of an in situ measurement associated with the calibration of a simulated DAA. The software simulates power measurements with Gaussian noise and assumes a priori knowledge of the initial phase control errors $\zeta_{km}$ and an ideal estimate of the initial amplitudes $\hat{A}'_k$ for each channel. The model allows the calibration procedure to be considered for both transmit and receive paths, and it is possible to set control errors in the paths using a random number generator with a given distribution law.

A linear DAA with the number of channels $N = 15$ was modeled. Using power measurement evaluation, estimates of the initial phase error in the calibrated channel are calculated. In the modeling results below, estimates were calculated in the seventh channel using $M$ samples of average power, and the number of samples can be varied. By setting the initial values of the emitted (received) power, depending on the type of calibration, as well as by changing the statistical characteristics of the noise, it is possible to set the values of the SNR and evaluate its influence on the calibration process and on the phase and amplitude errors in the calibrated channel.

To verify the validity of the simulation model, we first consider the case without control errors. If the SNR is such that estimate $\hat{\delta}_k = 0$, then according to the model, there are no phase errors in the calibrated path, as can be seen in Figure 17a. In the plots, the red line shows the variation of the absolute phase error $\delta_{abs}$, which characterizes the process of adding a phase shift during the simulation. The blue line shows the dynamics of the relative error $\hat{\delta}_k^{(i)} - \hat{\delta}_k^{(i-1)}$, which illustrates the process of completing the calibration with a given accuracy.

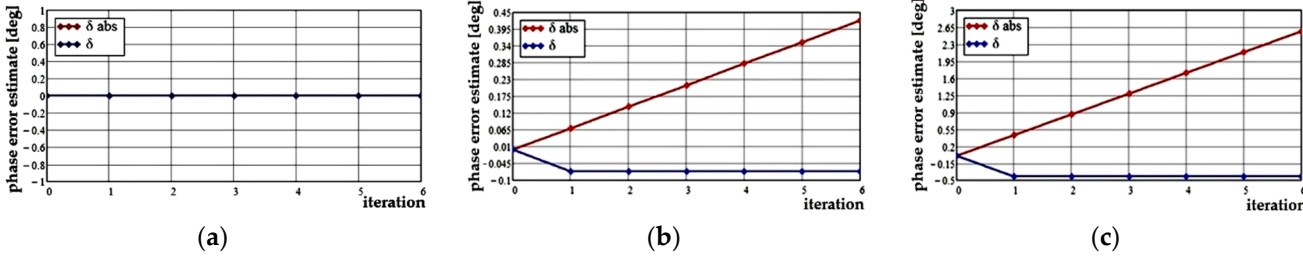

**Figure 17.** Convergence of the calibration process with 30 dB SNR: (**a**) $\zeta = 0$; (**b**) $\zeta \neq 0$ in one channel; (**c**) $\zeta \neq 0$ in six channels.

When a random phase control error $\zeta_{k0} = \zeta$ appears in any of the antenna array channels (see Figure 17b), the algorithm determines an estimate of the related phase error and processes it in one iteration, assuming for simplicity that the phase control errors are identical in all phase states. This approximation accurately reflects the situation when

calibrating the DAA receiving path, where this error is determined by the ADC jitter and does not depend on the value of the phase shift, i.e., sampling time. The same reasoning is valid for calibration of the transmission path, where the source of amplitude and phase control errors is a quadrature modulator in each digital transmission module. The process of working out the phase error in the model is stable; the estimate value does not change starting from the second iteration and is less than $0.1°$.

When the same phase errors appear in six channels out of fifteen (see Figure 17c), the process is repeated, but the phase error estimate is about $0.5°$, indicating that the calibration algorithm fully tracks the phase errors introduced using the control devices.

The model allows us to estimate the joint influence on the phase error of the SNR in the received (transmitted) signal and the number of samples in the calibrated channel. A comparison of the convergence processes of the calibration procedure in the simulation model is shown in Figure 18a,b. With the same number of samples $M = 4$, the model in Figure 18a where SNR is equal to 8 dB gives an initial estimate of the phase error with an accuracy of $\hat{\delta}_k^{(0)} = 6.5°$. If we increase the signal-to-noise ratio up to 16 dB with the same number of samples, the initial error estimate decreases to $\hat{\delta}_k^{(0)} = -2.25°$; the sign is irrelevant in this case, but shows that not only the absolute value of the error is determined, but also the phase shift direction. Increasing the number of signal samples in a calibration procedure to $M = 16$ while maintaining SNR equal to 16 dB (see Figure 18d) reduces the phase error of the calibration to $\hat{\delta}_k^{(0)} = 1.5°$. The convergence process in Figure 18c shows that increasing the signal-to-noise ratio from the case in Figure 18a to 10 dB and the number of samples to $M = 8$ halves the calibration error to $\hat{\delta}_k^{(0)} = -3.2°$.

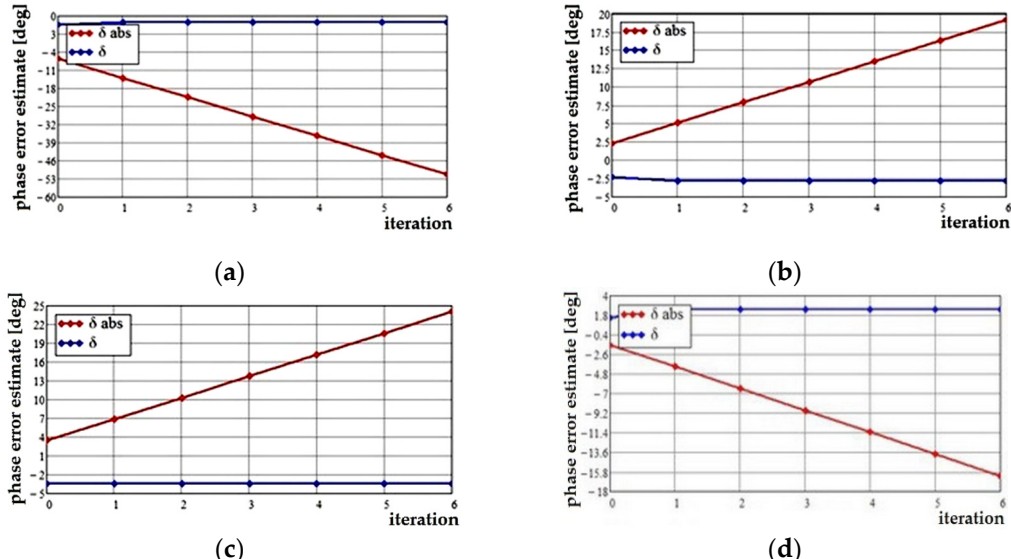

**Figure 18.** Examples of convergence of the calibration process: (**a**) SNR = 8 dB, $M = 4$, $\hat{\delta}_k^{(0)} = 6.5°$; (**b**) SNR = 16 dB, $M = 4$, $\hat{\delta}_k^{(0)} = -2.25°$; (**c**) SNR = 10 dB, $M = 8$, $\hat{\delta}_k^{(0)} = -3.2°$; (**d**) SNR = 16 dB, $M = 16$, $\hat{\delta}_k^{(0)} = 1.5°$.

For all cases presented in Figure 18, it can be noted that the process of determining the phase error converges with some accuracy, and this process is uniform. The error is reduced to a value determined by the error estimate corresponding to the specified system parameters. The special case where there is the initial phase shift in the path before the start of calibration $\hat{\delta}_k^{(0)} = \pm 180°$ is shown in Figure 19.

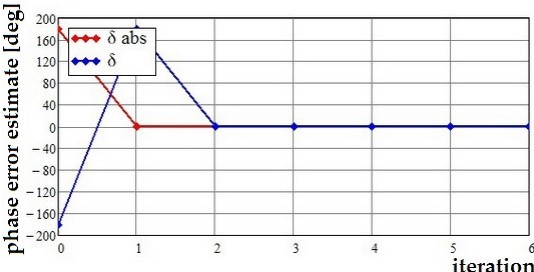

**Figure 19.** Convergence of the calibration process, $\hat{\delta}_k^{(0)} = 180°$.

This may cause the power value corresponding to the $\Phi_0$ phase state to become very small and, conversely, large in the $\Phi_{180}$ phase state, which can interfere with the calibration. However, the model shows that the process is delayed by only one iteration and after a phase jump in the channel to $180°$ at the second iteration, already at the third iteration the process is established and the error is clearly calculated.

Figure 20 allows us to evaluate the decisive influence of the signal-to-noise ratio on the convergence of the calibration process and on its stability. Figure 20a demonstrates the calibration procedure in a channel with SNR = 4 dB and the power is estimated from just one signal sample, i.e., $M = 1$. Phase and amplitude control errors in this calculation were equal to zero. The result of such initial data is a high phase error, $\hat{\delta}_k^{(0)} = 18°$, and the possibility of an unstable calibration, which is reflected in a jumping phase change after four convergence iterations. Looking at the iteration process over a larger time interval (see Figure 20b), we can see that such phase "spikes" occur periodically at every fifth iteration, but for the rest of the time the calibration process converges.

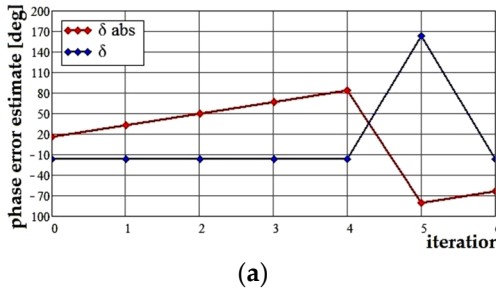

(**a**)

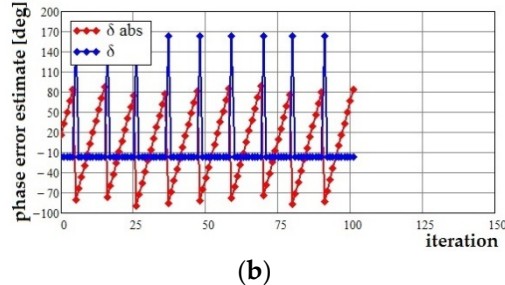

(**b**)

**Figure 20.** Convergence of the calibration process with SNR = 4 dB: (**a**) five iterations; (**b**) one hundred iterations.

This calibration method is suitable if there are strict time constraints for its implementation or if there is an antenna system in which it is impossible to improve the SNR using special methods.

The modeling of the DAA channel calibration process has revealed an important property of the proposed implementation of calibration, which is shown in a simplified form in Figure 21. An analysis of the convergence processes for various parameters of the calibrated system and the SNR shows that the influence of control errors is similar to the SNR impact. For example, in the case where the SNR is high (equals to 10 dB) and the number of samples for power estimation is also quite high, $M = 8$, the estimate of the initial phase error in the calibrated channel, excluding control errors, is $\hat{\delta}_k^{(0)} = 3.2°$, as can be seen in Figure 21b for the zero iteration. If a random control error with a possible maximum value of $\zeta \leq 45°$ is introduced into the model, which is quite possible in analog phase shifters, then the behavior of the calibration system becomes similar to the situation with the SNR being small, and the power is estimated using only one signal sample ($M = 1$), which corresponds to the case shown in Figure 21a. If the system parameters in the model remain the same, but without a control error ($\zeta = 0°$), then the convergence process

proceeds uniformly and initially the phase error estimate is determined stably, as shown in Figure 21b. If a control error appears uniformly distributed between all antenna array channels and its maximum standard deviation $\zeta \leq 8°$ (see Figure 21c), then in the zero iteration, this error is not tracked and amounts to $\hat{\delta}_k^{(0)} = 3.2°$, as for the case in Figure 21b; however, already at the first iteration the additional error is calculated, and the convergence process is completed by the second iteration.

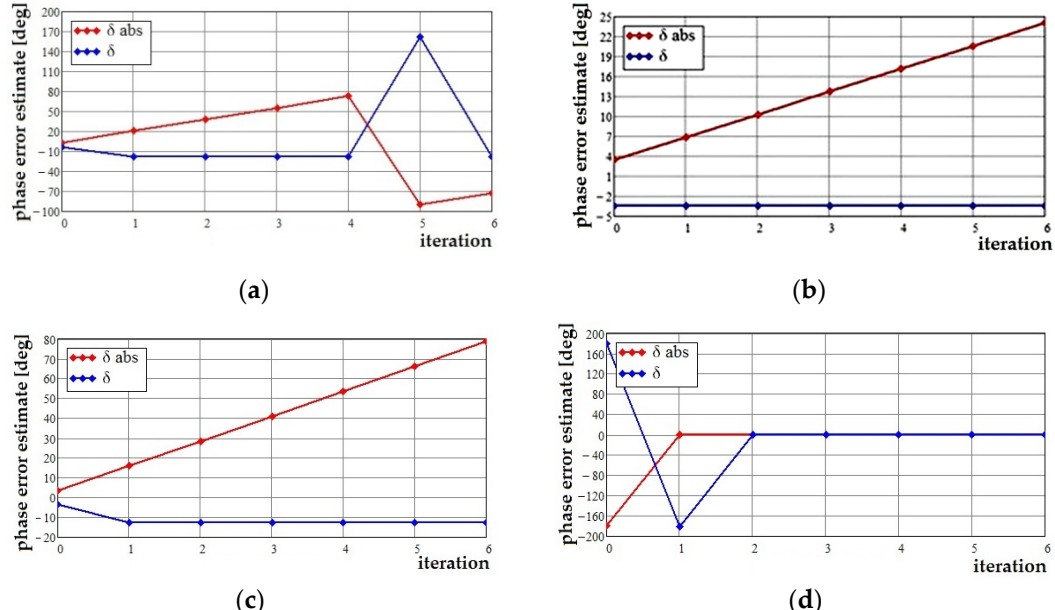

**Figure 21.** Examples of convergence of the calibration process: (**a**) SNR = 10 dB, $M = 8$, $\hat{\delta}_k^{(0)} = 3.2°$, $\zeta \leq 45°$; (**b**) SNR = 10 dB, $M = 8$, $\hat{\delta}_k^{(0)} = 3.2°$, $\zeta = 0°$; (**c**) SNR = 10 dB, $M = 8$, $\hat{\delta}_k^{(0)} = 3.2°$, $\zeta \leq 8°$; (**d**) SNR < 0 dB, $M = 1$, $\hat{\delta}_k^{(0)} = 180°$, $\zeta = 0°$.

The situation is similar when the signal strength is significantly lower than the noise level, SNR < 0 dB, which corresponds to the calibration procedure of the radar system under artificial interference conditions. If the number of samples is set to $M = 1$ to simulate the worst-case scenario, the initial phase error is very large, $\hat{\delta}_k^{(0)} = 180°$, even in the absence of control errors $\zeta = 0°$, as shown in Figure 21d. However, after two iterations, the error is determined, and the convergence process is stable until the sixth iteration.

In all the examples given, the method differentially converges in a few iterations, and it is assumed that the parameters of the DAA channels and their electronic components change slightly during the calibration process.

### 3.2. Results of Experimental Studies of Calibrating Algorithms

The measurement setup for validating the simulation model of the amplitude–phase distribution diagnostic procedure presented in Section 2 was assembled in an anechoic chamber. That setup enables replicating the calibration procedure using the proposed method.

The calibrated DAA was simulated by sequentially moving the measurement antenna along the X-axis of the planar scanner, as described in Section 3. The measurements were performed at a 10 GHz carrier frequency, with the antenna moving in steps of $\lambda/2 = 15$ mm over a distance of 210 mm, taking $N = 15$ discrete positions on the planar positioner. The far-field distance for an antenna with similar parameters is 2.94 m, while in the experiment the calibration signal transmitter was placed at a distance of more than 3.25 m from the scanner. For the proposed DAA calibration method, the angular position of the calibration antenna is insignificant, but it must be well known. However, to minimize instrumental

errors, it was installed normal to the plane of probe movement along the scanner during the experiment.

For the purposes of verifying the diagnostic algorithm experimentally, a single position of the probe on the scanner was chosen, corresponding to the seventh channel, as in the simulation model. The diagnostic procedure was carried out in two ways. First, to simulate the introduction of a phase shift, the probe was moved along the Z-axis of the scanner to the corresponding distance. For each phase state, 16 samples of the complex signal were measured. In the second case, various time samples of the signal in the calibrated channel were used. The required four phase states were introduced mathematically. The sources of amplitude errors in the experiment were internal reflections of the anechoic chamber and its components. The number of reflective objects was varied to replicate different SNR at the input of the calibrated element. The sources of phase errors were the positioning accuracy of the scanner and the intentional deviation of the ideal linear movement during antenna array simulation as well as errors included by roughly moving the probe along the Z-axis to achieve required phase states during the diagnostic process.

All measured values were loaded into a mathematical model, in which the necessary calculations were performed and the results are presented graphically.

The conditions were reproduced that are extremely consistent with the initial simulation ones, and characteristics similar to those given in Section 4 were constructed. Figure 22a shows the convergence of the calibration procedure in the selected seventh channel when introducing a phase shift by the Z-axis movement. The situation of no control errors, corresponding to the estimation of channel errors using (3) and (4), was reproduced with precise preliminary adjustment. Assuming $SNR = 30$ dB during the experiment, which follows from the estimate of the RMS phase error of calibration $\sigma_\delta^2 = 0$ (see Figure 17a), the results obtained for the calibrated channel are $\delta_7 = 4.47°$ and $A_k = -32.4$ dB. The phase error in the calibrated channel corresponds to the deviation of the antenna position on the scanner from the specified value by $\Delta L = 0.37$ mm. The detection of such errors characterizes the high accuracy of the proposed algorithm. The tracking of the model's phase error is stable, which can be seen in Figure 22b. The error estimation does not change starting from the second iteration and is equal to $0°$.

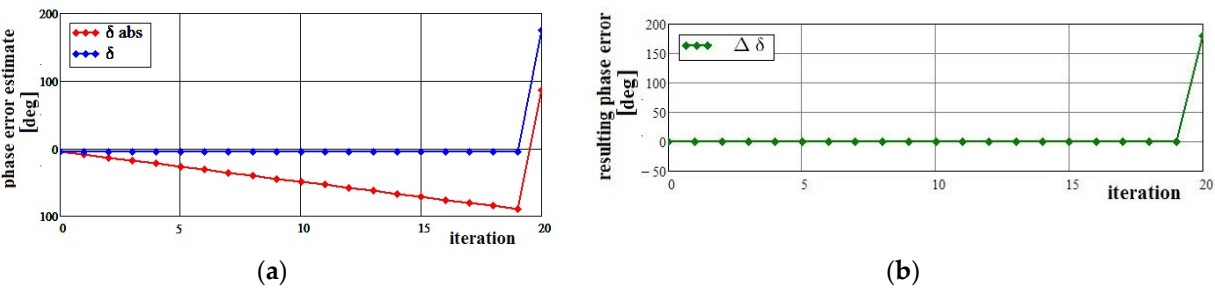

**(a)**　　　　　　　　　　　　　　　　　　　　**(b)**

**Figure 22.** Convergence of the calibration process for an experimental DAA at 10 GHz carrier frequency: (**a**) $\zeta = 0$; (**b**) the resulting phase error.

When adding an artificial phase control error in the range from $9°$ to $18°$ for five of the fifteen channels, the algorithm determines an estimate of the total phase error in the calibrated channel $\hat{\delta}_7 = 18.9°$ (see Figure 23a) and processes it in one iteration (see Figure 23b).

A slight reduction in SNR, by reducing the radiated power and increasing the VNA's RBW, with the same phase control error in the same channels, leads to an increase in the phase deviation in the calibrated channel to $23.18°$, and the relative amplitude error becomes equal to $-28$ dB. The kind of dependencies illustrating the convergence of the calibration algorithm do not change significantly.

While using different time samples of the signal in the calibrated channel and mathematically calculating the required four phase states, reflections in the AEC were artificially

increased by placing reflective objects, the number and location of which was selected to simulate various SNR at the input of the calibrated channel. The sources of phase errors were now only the accuracy of the scanner positioning, and errors introduced by rough movement of the probe along the Z-axis were excluded [64]. The results of the phase error estimation for SNR = 16 dB are shown in Figure 24.

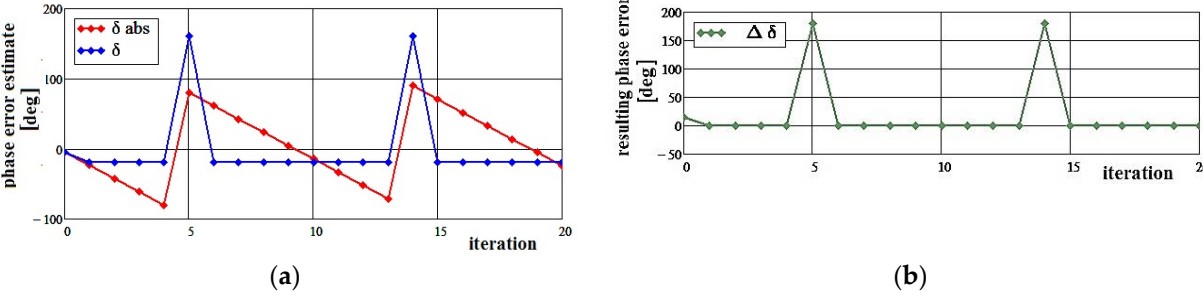

(**a**)         (**b**)

**Figure 23.** Convergence of the calibration process for an experimental DAA at 10 GHz carrier frequency: (**a**) $\zeta \neq 0$ for five channels; (**b**) the resulting phase error.

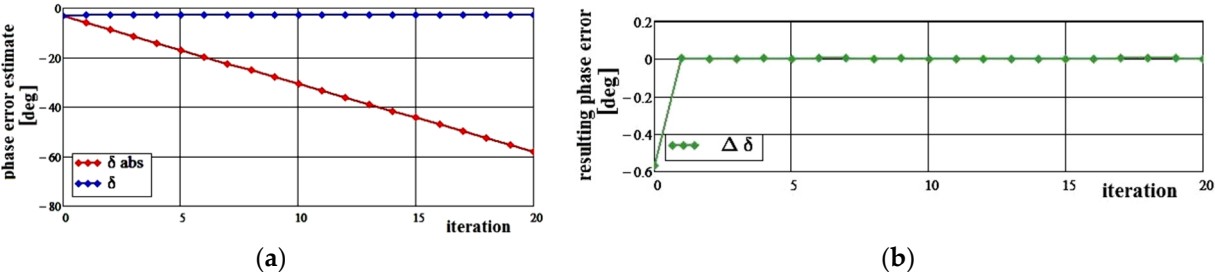

(**a**)         (**b**)

**Figure 24.** Calibration process with mathematical calculation of phase states: (**a**) Convergence of the calibration process with instrumental errors in five channels; (**b**) the resulting phase error.

The transition from manually introducing the required phase shifts to a mathematical one reduces the total phase error in the calibrated channel to 2.7°, with an initial error of 0.5°, as can be seen in Figure 24a. Reducing the SNR increased the relative amplitude error to −28 dB.

For all cases presented in Figures 22–24, it can be noted that the process of determining the phase error uniformly converges with high accuracy, and the error decreases to the minimum possible value, determined with the specified system parameters and calculation methods.

## 4. Discussion

A comparison of the results for the studied algorithm for channel-by-channel calibration of the DAA for various values of SNR and the number of signal samples in the calibrated channel $M$ is given in Table 2.

The advantage of the proposed algorithm is its convergence under most unfavorable initial conditions. Increasing the SNR is the most effective way for reducing the initial phase error of the calibration. If common operating conditions do not allow for an increase in the SNR, then it is necessary to increase the number of signal samples $M$ in the calibrated channel. The number of samples has less of an impact on the accuracy of the algorithm, while the time required for calibration will inevitably increase.

The numerical results of the influence of control errors in channels $\zeta$ on the initial phase calibration error are summarized in Table 3. It can be seen that the influence of control errors is similar to a decrease in the SNR. As the control errors' $\zeta$ increases and the SNR decreases in the channel, the process of determining errors is delayed. If SNR ≤ 0 and the number of samples $M = 1$, then this leads to instability of the calibration process. However,

this type of calibration may be acceptable when time resources for its implementation are limited and the calibration is stopped just before phase "jumps".

**Table 2.** The influence of SNR on the initial phase error of calibration.

| SNR, dB | $M$ | $\hat{\delta}_k^{(0)}$, Degree | Convergence of the Calibration Process | Possible Instability of the Calibration Process |
|---------|-----|-------------------------------|----------------------------------------|------------------------------------------------|
| $\leq 0$ | 1 | 180 | + | + |
| 4 | 1 | 18 | + | + |
| 8 | 4 | 6.5 | + | − |
| 10 | 8 | −3.2 | + | − |
| 16 | 4 | −2.25 | + | − |
| | 16 | 1.5 | + | − |

**Table 3.** Effect of control errors on the convergence of the process.

| SNR, dB | $M$ | $\hat{\delta}_k^{(0)}$, Degree | $\zeta$, Degree | At Which Iteration the Error Is Detected | At Which Iteration Is the Instability Observed |
|---------|-----|-------------------------------|-----------------|------------------------------------------|-----------------------------------------------|
| $\leq 0$ | 1 | 180 | 0 | 2 | 5 |
| 4 | 1 | 18 | 0 | 0 | 5 |
| 10 | 8 | 3.2 | $\leq 45$ | 1 | |
| | | | $\leq 8$ | 1 | stable |
| | | | 0 | 0 | |

The results of the experiment with a DAA simulation model, presented in Table 4, show that the use of special mathematical processing in antenna channels, excluding physical changes in phase shifts in the calibrated channel, can significantly reduce the phase error even with a decrease in SNR. The $\Delta L$ parameter is calculated as the change in the microwave path length, equivalent to the determined phase error. For example, $\Delta L = 0.22$ mm at an operating frequency of 10 GHz demonstrates a significant improvement in accuracy compared to traditional analog calibration methods [50].

**Table 4.** Experimental results.

| Phase Shift | SNR | $\zeta$, Degree | $\hat{\delta}_7$, Degree | $\hat{A}_k$, dB | $\Delta L$, mm |
|-------------|-----|-----------------|--------------------------|-----------------|----------------|
| Physical antenna moving | 30 | 0 | 4.47 | −32.4 | 0.37 |
| | | 9…18 in 5 channels out of 15 | 18.9 | −31 | 1.6 |
| Mathematically introduced | 16 | 9…18 in 5 channels out of 15 | 2.7 | −28 | 0.22 |

In future work, it is planned to create a prototype of a radio-electronic system with DAA and use the investigated algorithms to achieve high accuracy of APD over its aperture.

## 5. Conclusions

1. The implementation of the calibration method for an on-board multi-beam digital antenna array is proposed. The advantage of this scheme is that a number of analog devices that introduce uncontrolled random amplitude and phase errors into the beamforming process are excluded. In perspective, this solution could increase the pointing accuracy of each DAA beam to fractions of a degree.

2. The scalar calibration algorithm is modified for digital antennas. Its accuracy characteristics have been studied: high performance and convergence of the algorithm for the corresponding SNR ratios and sample sizes are obtained.

3. A series of experiments were carried out in an anechoic chamber using an antenna mounted on a planar scanner, as a result of which the sum and the difference digital radiation patterns of the DAA simulation model were constructed. The direction finding procedure, that is, the direction of arrival, was estimated with high accuracy.

4. The possibility of calibrating the DAA during its main operation with accuracies exceeding known analog systems has been experimentally shown.

5. Theoretical provisions on the possibility of diagnosing an antenna array with an estimate of the initial phase error $\hat{\delta}_k^{(0)} = 0.5°$ have been proved.

The presented algorithm is promising and can also be considered as a diagnostic tool for microwave paths in various radio-electronic systems. It is suitable for newly developed and applied on-board DAAs, as well as for operation as part of modern telecommunication systems [65]. In the future, it is planned to make a prototype of a digital antenna array to test digital signal processing techniques and study its electrical characteristics. The implementation of super-resolution algorithms in DAA and the creation of conformal antenna systems for integration with the surface of the carrier also seem promising for further research. The effectiveness of work in these areas will be determined by progress in improving the technology of the digital element base, increasing its performance while reducing weight, size, and cost.

**Author Contributions:** Conceptualization, E.D. and M.S.; methodology, E.D.; software, M.S. and V.S.; validation, V.S. and T.S.; formal analysis, E.D. and T.S.; investigation, M.S. and T.S.; resources, M.S. and V.S.; data curation, V.S.; writing—original draft preparation, E.D. and M.S.; writing—review and editing, M.S. and T.S.; visualization, M.S. and V.S.; supervision, E.D.; project administration, T.S.; funding acquisition, E.D. All authors have read and agreed to the published version of the manuscript.

**Funding:** This research was funded by state assignment of the Ministry of Science and Higher Education of the Russian Federation, project no. FSFF-2023-0005.

**Institutional Review Board Statement:** Not applicable.

**Informed Consent Statement:** Not applicable.

**Data Availability Statement:** Data are contained within the article.

**Conflicts of Interest:** The authors declare no conflict of interest.

## Abbreviations

The following abbreviations are used in this manuscript:

| | |
|---|---|
| ADC | Analog-to-Digital Converter |
| AEC | Anechoic Chamber |
| APD | Amplitude–Phase Distribution |
| CCF | Cross-Correlation Function |
| DAC | Digital-to-Analog Converter |
| DAA | Digital Antenna Array |
| DFT | Discrete Fourier Transform |
| DTRM | Digital Transceiver Module |
| DoA | Direction of Arrival |
| DSP | Digital Signal Processing |
| DSPC | DSP at Calibrated Channel |
| DRP | Digital Radiation Pattern |
| IDFT | Inverse Discrete Fourier Transform |
| MIMO | Multiple-Input Multiple-Output |
| PAA | Phased Antenna Arrays |
| RBW | Resolution Bandwidth |
| RES | Radio-Electronic System |

| REV | Rotating Element Electric Field Vector |
|-----|----------------------------------------|
| RMS | Root Mean Square |
| SLL | Sidelobe Level |
| SNR | Signal-to-Noise Ratio |
| TRM | Transceiver Module |
| UAV | Unmanned Aerial Vehicle |
| VNA | Vector Network Analyzer |

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
