# Peer review of "Development of the Phaseless Calibration Algorithm for a Digital Antenna Array"

_inventions, doi:10.3390/inventions8060155_

Round 1

Reviewer 1 Report

Comments and Suggestions for Authors

In this manuscript, the authors introduce a new phaseless calibration algorithm for digital antenna array. The proposed algorithm can be applied for performance diagnostic during the whole antenna system life cycle. The topic is interesting and worth studying. The manuscript is well- organized and well-written. The simulation results show the effectiveness of the proposed algorithms in terms of the convergence of the calibration process and resulting phase error.

My remarks are as follows:

1. In the Discussion section, include a comparison with the results obtained from previous similar studies that were mentioned in the “1. Introduction” section. Highlight the advantages and disadvantages of the proposed approach.

2. In the Conclusion section, study limitations and future research are not mentioned.

3. Provide more details about your software implementation in terms of the development environment, programming language, and libraries used. Consider including a link to the source code for reference.

Author Response

First of all, we would like to thank the Reviewer for their careful review of our paper and their comments which have helped us improve our paper.

In this revised version of paper, we have made necessary change to address all comments raised by all the Reviewers. The main changes are the following. Firstly, we have added a summary table of the calibration methods mentioned in the Introduction; secondly, we have improved the description of further directions of the development of our research.

  • Discussion section

In the Discussion section, include a comparison with the results obtained from previous similar studies that were mentioned in the “1. Introduction” section. Highlight the advantages and disadvantages of the proposed approach.

The recommended comparisons have been carried out and included in the Introduction section. The analysis of antenna array calibration methods is summarized in the corresponding table (Table 1).

  • Conclusion section

In the Conclusion section, study limitations and future research are not mentioned.

In the Conclusion section, we have added a paragraph describing future directions for the work.

  • Software implementation

Provide more details about your software implementation in terms of the development environment, programming language, and libraries used. Consider including a link to the source code for reference.

Writing software was not the main goal of the work, so we do not focus on it. The software in this study performs purely computational and graphical functions and does not contain any new results. It serves only to speed up work with the datasets and conveniently performs the necessary calculations. Therefore, the authors consider it reasonable not to divert the readers’ attention with such an unimportant detail. However, for the sake of clarification, we would like to mention that the software was implemented in the Scilab software environment.

Reviewer 2 Report

Comments and Suggestions for Authors

In the present paper, the phaseless calibration algorithm is studied in a set of digital antennas.The scalar method has been used for multibeam digital antennas, thus an improvement in orientation is expected by reducing errors in the reception/transmission channels, improving the angle of arrival through electronic scanning.

It is a very well presented work that is easy to follow.

It must be corrected:

Figure 17 second b) must be c)

Line 470: Figure 2?

Author Response

First of all, we would like to thank the Reviewer for their careful review of our paper. We highly appreciate your positive opinion about our research work.

In this revised version of paper, we have done necessary changes to address all comments raised by the Reviewers, in particular, (i) we have corrected typos in the figure captions and (ii) we have updated the opening paragraph in subsection 3.2.

  • Figures

It must be corrected:

Figure 17 second b) must be c)

We made corrections the revised version of the document.

Line 470: Figure 2?

In the edited version of the article we have referred to Section 2 instead of the Figure 2.

Reviewer 3 Report

Comments and Suggestions for Authors

A list including all the many used acronyms should be provided for its easy identification as they appear in the text when reading the paper.

A new part of the last paragraph of the introduction or close new paragraph should describe clearly what is specifically new in this research.

Some basic formulas as, for instance, (2) or (3)-to (6),  need further explanations and, probably , explicit  citation from  “ ad -hoc” references.

The subsections of Section 2 and 3 should be split into more sub-subsections to identify clearly when reading the various concerns in the simulations and experimental implementations.

It is not clear when reading the distinction between the experimental calibration method and the simulation model. In other words, is it experimentally corroborated what the simulation model predicts?. For instance, is Eqn. (2) implemented through some physical experiment?.

Eqn. 2 is not clearly explained. Why the convolution is performed with signals emitted  from different positions?.

Eqns. (5)-(6) are referred to the explanation in the former text  paragraph. The last full-stop in line 336, page 12, should be replaced with “:”. It is not explained how (5)-(6) are obtained.

Author Response

First of all, we would like to thank the Reviewer for their careful review of our paper and their comments which have helped us improve our paper.

In this revised version of paper, we have made necessary changes to address all the comments raised by the Reviewers, in particular:

(i) we added acronym section in the end of the article;

(ii) we have improved the article structure, clarifying the novelty of the work;

(iii) we have enhanced the paper outlook by extending clarification to the basic equations;

(iv) we added missing references to the digital processing techniques we used;

(v) we have clarified the “simulation model” term which was used.

  • Abbreviations

A list including all the many used acronyms should be provided for its easy identification as they appear in the text when reading the paper.

We have added the corresponding section to the article.

  • Novelty

A new part of the last paragraph of the introduction or close new paragraph should describe clearly what is specifically new in this research.

We have added corresponding revisions to the article.

Clarification

Let us combine a number of the reviewer’s notes.

  1. Some basic formulas as, for instance, (2) or (3)-to (6), need further explanations and, probably, explicit citation from “ad-hoc” references. Eqn. 2 is not clearly explained. Why the convolution is performed with signals emitted from different positions? Eqns. (5)-(6) are referred to the explanation in the former text paragraph. The last full-stop in line 336, page 12, should be replaced with “:”. It is not explained how (5)-(6) are obtained.

For eqn. (2), we had given poorly explanations in the initial submission. Basically, we have carefully reviewed this part of the paper and made corrections in the revised version of the document. The expressions we use are taken from respectful publications. For example, the fast convolution method implemented in eqn. (2) is considered to be the main algorithm for digital beamforming. It is described rather well in a large number of research papers and even some recent teaching book. We added the following references to our survey:

[63]. Oppenheim A.V.; Schafer R.W., Discrete-Time Signal Processing, 3rd ed. Prentice Hall Signal Processing, 2009.

[64]. Falk J., Händel P., Jansson M., Direction Finding for Electronic Warfare Systems Using the Phase of the Cross Spectral Density. in RadioVetenskap och Kommunikation (RVK), 2002, pp. 264–268.

Equations (3)-(6) are well known and relate to the research of R. Sorace, a specialist in the field of PAA calibration. We have added explicit references to these equations in the preceding paragraph, thus, reference [61] is the patent in which some similar yet basic ideas related to the calibration method we exploit could have been proposed for the first time. In addition, at the beginning of Section 2.1, we have added reference [58] to the paper written by the same author, where these relationships are discussed in detail. The novelty that may stand our research out is that we practically examined at the measurement site how applicable similar approach for DAA as we practically tested the algorithm for finding errors using newer version of the technique.

  1. The subsections of Section 2 and 3 should be split into more sub-subsections to identify clearly when reading the various concerns in the simulations and experimental implementations.

In the article, Sections 2 and 3 are already divided into subsections corresponding to the topic. We try avoiding excessive partitioning in order to fulfill the demand for structure in Inventions journal.

  1. It is not clear when reading the distinction between the experimental calibration method and the simulation model. In other words, is it experimentally corroborated what the simulation model predicts? For instance, is Eqn. (2) implemented through some physical experiment?

In the presented work, we understand the simulation model as a physical model of the antenna array obtained where a single emitter is being moved. In other words, it is a sort of aperture synthesis. Furthermore, mathematical modeling occurs when the estimation of the direction of arrival is performed. At that point, it is necessary to obtain the parameters of the received signal coming from the direction of interest during the procedure based on cross-correlation evaluation. We have made appropriate clarifications in the article.

This paper uses a phaseless switching method for antenna array calibration. The developed software that realizes the calibration algorithm is not a calibration model. Thus, there is no difference between the experimental method and the simulation model. Eqn. (2) is used to post-process the experimental data to mathematically represent the digital beamforming in the DAA model. This mathematical apparatus is used in real DAAs.

Reviewer 4 Report

Comments and Suggestions for Authors

The manuscript is interesting and seems to contain novel ideas. Please find below some parts that still need to be improved:

1) The beginning of the Introduction does not motivate properly the actual contribution of the manuscript. It starts with discussions on UAV technologies, which however represent only a small part of the more important ecosystem of telecommunication technologies that would benefit from digital beamforming and online calibration. I would suggest to significantly modify this part and offering a wider discussion on the potential of DAA in current and emerging telecommunication systems. From a quick search on Google Scholar, some interesting examples that justify the need of enhanced online calibration capabilities for real-time digital beamforming in emerging 5G and beyond systems can be found, as for instance "RIS-aided joint localization and synchronization with a single-antenna receiver: Beamforming design and low-complexity estimation", IEEE JSTSP, 2022, but also "Statistical digital predistortion of 5G millimeter-wave RF beamforming transmitter under random amplitude variations", IEEE TMTT, 2022. In this reviewer's opinion, this part needs to be greatly improved as it lacks most of the main motivating literature.

2) At the end of the Introduction, please provide a Table summarizing the main pros and cons of the discussed (existing) calibration techniques, so as to more clearly highlight their main limitations on how the present contribution helps in advancing the state-of-the-art.

3) Please use the same green color for Figures 22b and 23b.

4) Is it possible to test DOA estimation using a more suitable estimation algorithm such as the MUSIC, instead of searching for the maximum of the CCF? Moreover, it would be interesting to evaluate more challenging estimation conditions such as signals impinging close to the end-fire of the array.  

5) The possible future research directions should be better discussed in the Conclusion section.

Comments on the Quality of English Language

N/A

Author Response

First of all, we would like to thank the Reviewer for their careful review of our paper and their comments which have helped us improve our paper.

In this revised version of paper, we have made necessary corrections to address all comments raised by the Reviewers, in particular:

(i) we have added a summary table of the calibration methods mentioned;

(ii) we have enhanced the paper outlook by aligning the colour scheme used for the figures;

(iii) we have improved the description of further areas of work.

  • Application

The beginning of the Introduction does not motivate properly the actual contribution of the manuscript. It starts with discussions on UAV technologies, which however represent only a small part of the more important ecosystem of telecommunication technologies that would benefit from digital beamforming and online calibration. I would suggest to significantly modify this part and offering a wider discussion on the potential of DAA in current and emerging telecommunication systems. From a quick search on Google Scholar, some interesting examples that justify the need of enhanced online calibration capabilities for real-time digital beamforming in emerging 5G and beyond systems can be found, as for instance "RIS-aided joint localization and synchronization with a single-antenna receiver: Beamforming design and low-complexity estimation", IEEE JSTSP, 2022, but also "Statistical digital predistortion of 5G millimeter-wave RF beamforming transmitter under random amplitude variations", IEEE TMTT, 2022. In this reviewer's opinion, this part needs to be greatly improved as it lacks most of the main motivating literature.

Telecommunication systems (TCS) have their own specific features, and one of possible application of our work is this field. But the authors' current research interest is mostly focused in the problems of calibration of DAA placed on unmanned aerial vehicles, i.e., where there is no easy and/or direct access to DAA channels. However, we have added in the Conclusion a suggestion about the applicability of the methodology in tasks related typically to TCS.

  • Review summary

At the end of the Introduction, please provide a Table summarizing the main pros and cons of the discussed (existing) calibration techniques, so as to more clearly highlight their main limitations on how the present contribution helps in advancing the state-of-the-art.

We have made related corrections in the revised version of the document.

  • Figures

Please use the same green color for Figures 22b and 23b.

We have made related corrections in the revised version of the document.

  • Algorithms

Is it possible to test DOA estimation using a more suitable estimation algorithm such as the MUSIC, instead of searching for the maximum of the CCF? Moreover, it would be interesting to evaluate more challenging estimation conditions such as signals impinging close to the end-fire of the array.

The main goal of our current work is to study and experimentally confirm the considered calibration algorithm for phased array antennas in relation to digital ones. Section 2 is devoted to the synthesis of the antenna array model. For digital signal processing technique, such as beamforming of the receiving radiation pattern, the simplest method to implement was actually chosen for determining angular coordinates. Nevertheless, each direction finding algorithms has its own limitations and disadvantages, therefore, the choice of the most suitable algorithm mostly depends on the purpose of DAA, its physical geometry and other factors, the study of which was out of scope of our work. However, many practitioners working with real direction-finding systems claim that the accuracy of angular coordinate estimation will always decrease due to the presence of unaccounted amplitude-phase errors regardless of the chosen algorithm. In other words, we are sure that the reducing this sort of errors will definitely be crucial for the system implemented any particular algorithm.

  • Conclusion section

The possible future research directions should be better discussed in the Conclusion section.

In the Conclusion section, we have added a paragraph describing future directions for our work.

Round 2

Reviewer 4 Report

Comments and Suggestions for Authors

The authors have correctly addressed all my comments.

Comments on the Quality of English Language

N/A